# The Discovery, Enzymatic Characterization and Functional Analysis of a Newly Isolated Chitinase from Marine-Derived Fungus *Aspergillus fumigatus* df347

**DOI:** 10.3390/md20080520

**Published:** 2022-08-15

**Authors:** Ya-Li Wu, Sheng Wang, Deng-Feng Yang, Li-Yan Yang, Qing-Yan Wang, Jun Yu, Nan Li, Li-Xia Pan

**Affiliations:** 1College of Life Science and Technology, Guangxi University, Nanning 530004, China; 2State Key Laboratory of Non-Food Biomass and Enzyme Technology, Guangxi Key Laboratory of Marine Natural Products and Combinatorial Biosynthesis Chemistry, Guangxi Academy of Sciences, 98 Daling Road, Nanning 530007, China; 3Nanning Pangbo Biological Engineering Co., Ltd., Nanning 530004, China; 4College of Food and Quality Engineering, Nanning University, Nanning 530200, China

**Keywords:** chitinase, strain screening, enzymatic properties, chitin oligosaccharides, heterologous expression, molecular docking

## Abstract

In order to discover a broad-specificity and high stability chitinase, a marine fungus, *Aspergillus fumigatus* df347, was identified in the sediments of mangrove wetlands in Qinzhou Bay, China. The chitinase gene (*Af*Chi28) from *A. fumigatus* df347 was cloned and heterologously expressed in *Escherichia coli*, and the recombinant enzyme *Af*Chi28 was purified and characterized. *Af*Chi28 is an acido-halotolerant- and temperature-resistant bifunctional enzyme with both endo- and exo-cleavage functions. Its enzymatic products are mainly GlcNAc, (GlcNAc)_2_, (GlcNAc)_3_ and (GlcNAc)_4_. Na^+^, Mg^2+^, K^+^, Ca^2+^ and Tris at a concentration of 50 mM had a strong stimulatory effect on *Af*Chi28. The crude enzyme and pure enzyme exhibited the highest specific activity of 0.737 mU/mg and 52.414 mU/mg towards colloidal chitin. The DxDxE motif at the end of strand *β*5 and with Glu154 as the catalytic residue was verified by the AlphaFold2 prediction and sequence alignment of homologous proteins. Moreover, the results of molecular docking showed that molecular modeling of chitohexaose was shown to bind to *Af*Chi28 in subsites −4 to +2 in the deep groove substrate-binding pocket. This study demonstrates that *Af*Chi28 is a promising chitinase for the preparation of desirable chitin oligosaccharides, and provides a foundation for elucidating the catalytic mechanism of chitinases from marine fungi.

## 1. Introduction

Chitin, linked by repeated β-1,4-*N*-acetylglucosamine (GlcNAc) units, is the second most abundant polysaccharide after cellulose. Chitin is mainly found in fungal, coralline algae, sponges, mollusks, insects and crustacean shells [1,2,3,4,5]. As an important biological resource, chitin is second only to cellulose in content on the earth, with an annual natural production of 10^14^ tons, which is a huge renewable resource. Approximately 10^12^–10^14^ tons of chitin are produced annually by living organisms in the ocean. Thus, it can be seen that chitin mainly comes from the ocean [6]. The amount of microbial decomposition of chitin in nature is very low, chitin waste accumulation leads to environmental and marine pollution, which also limits its potential applications in the fields of food, medicine, and agriculture [7,8,9,10].

Chitin derivatives are of great significance at present. The chitin oligosaccharides (*N*-acetyl COSs) are linked by β-1,4-glycosidic bonds. *N*-acetyl COSs of different chain lengths exhibit different functions, such as antitumor, immune stimulation and antioxidant activity [11,12]. *N*-acetyl COSs with the degree of polymerization (DP) ranging from two to ten, can be prepared by chemical, physical or enzyme processing methods [1,13]. The chemical production of chitin oligosaccharides is rapidly reactive, but the addition of organic solvents causes post-treatment difficulties and the low degree of polymerization of the hydrolysis product leads to GlcNAc being the main product [14,15,16,17]. Lin et al. [18] degraded chitin with nitrite to obtain chitin oligosaccharide products of different molecular weights, but the reaction process and the molecular weight distribution of the products were more difficult to control, the yield was low, and it was easy to cause pollution to the environment. Xing et al. [19], used a physical method (microwave irradiation) to hydrolyze chitin. Changing the hydrolysis time and microwave intensity obtained different molecular weight products, but the high cost is unsuitable for industrial production. Compared with the high-polluting chemical degradation and the high-energy-cost physical degradation, the production of *N*-acetyl COSs by the chitin enzymatic method has mild reaction conditions, high yields, low side reactions and minor pollution [20,21]. Chitinases play a key role in the biotransformation of chitin, by effectively catalyzing the cleavage of β-1,4-bonds in chitin, mainly releasing *N*-acetyl COSs [22,23,24]. Based on amino acid sequence similarities, chitinases are mainly distributed in three glycoside hydrolase (GH) families: GH18 [23,25], GH19 [26] and GH20 [27]. According to the type of cleavage, the chitinases are divided into endo-type chitinases (EC3.2.1.14) and exo-type chitinases (EC3.2.1.52) [28,29]. Xing et al. [30] utilized the broad-specific chitinases of *Penicillium oxalicum* k10 to degrade chitin to produce chitin oligosaccharides. However, they are not suitable for industrial applications due to their poor thermal stability. Some researchers try to use chitinases containing one or two carbohydrate-binding modules (CBM) to improve the efficient degradation of insoluble chitin. Nevertheless, compared with using the soluble chitin as a substrate by chitinases, the catalytic efficiency of this method is very low [31]. Interestingly, lytic polysaccharide monooxygenase (LPMO) oxidizes recalcitrant polysaccharides and promotes the conversion of chitin, by chitinase. Yang et al. [32] showed that LPMO CBP21 promoted the degradation of three chitin substrates (colloidal chitin, β-chitin and α-chitin) by *Aeromonas veronii* B565 chitinase Chi92. However, it produces a high level of aldonic acid as a byproduct. Therefore, many studies have attempted to discover new chitinases with better performance [33].

As the field of biotechnology continues to expand, there is a need to obtain chitinases with industrially applicable operational characteristics. Thermophilic chitinases are very important in recycling processes due to the rise in temperature which is an important phase during bioconversion of wastes. Enzymes from subtropical marine microorganisms are not only extremely salt-tolerant, but also heat-tolerant due to the specific environment in which they are found [34]. They seem to be good candidates for industrial applications.

So far, multiple kinds of chitinases have been reported from different sources, including plants, fungi and bacteria [30,35]. Among them, a chitinase from filamentous fungi has received more attention due to its advantages in the efficiency of chitin hydrolysis. Several species of the chitinase have been identified biochemically, such as chitinases from *A. oryzaeIn* [36], *P.oxalicum k10* [30]. It is well known that the genomes of *A. fumigatus* are rich in abundance of chitinase genes. However, most previous studies on *A. fumigatus* have focused on terrestrial organisms [37]. As Qinzhou Bay has a subtropical maritime monsoon climate, its harbor is rich in resources and has a pleasant climate [18,38]. It is one of the hotspots of biodiversity research in the world. The microbial community structure and diversity of Qinzhou Bay Harbor is very different from other regions, so it has attracted much research. In this study, a chitinase-producing strain was obtained from the sediment of the mangrove wetlands in the Guangxi Zhuang Autonomous Region and lives in the intertidal zone of the subtropical coast [39,40], which has the characteristics of a high salt (about 2% NaCl) and high heat environment. According to the results of an ITS rDNA sequence analysis, the screened strain was identified as *A. fumigatus* df347. The chitinase gene (*Af*Chi28) was mined from the cDNA of the strain and heterologously expressed in *Escherichia coli* BL21 codon plus (DE3) RIL. The recombinant enzyme *Af*Chi28 was purified and characterized. Its enzymatic properties and the degradation products were investigated, respectively. Furthermore, AlphaFold2 structure prediction and molecular docking were used to investigate the catalytic mechanism of chitinase from marine origin fungi.

## 2. Results and Discussion

### 2.1. Identification of ITS rDNA Sequence of the Strain

The single colony named df347, was isolated by the plate separation method and grew on the separation plate with chitin as the only carbon source, indicating that it has the ability to degrade chitin. The colony df347 was preserved in our laboratory. Sequence homology studies using the BLAST function on the NCBI website showed that the screened strains naturally clustered with the ITS rDNA sequences of *Aspergillus* sp. A phylogenetic tree was constructed using the software MEGA-X, as shown in Figure 1. It can be seen that the screened strains are up to 100% similar to *A. fumigatus Af*293 (XM 747819.1).

### 2.2. Sequence Analysis of AfChi28

The primers were designed based on the sequence of the genome of *A. fumigatus* Af293 from the NCBI database. The cDNA of *A. fumigatus* df347 was used as a template for PCR amplification of the target gene, which encodes a protein named *Af*Chi28. The CAZy annotation revealed that the protein *Af*Chi28 contains a GH18 domain. The theoretical molecular mass of *Af*Chi28 is 40.9 kDa with a pI of 7.03, which was computed by the ExPASy ProtParam tool. The signal peptide of *Af*Chi28 was calculated by the SignalP-5.0, and the result showed *Af*Chi28 may not contain a sec signal peptide. SMART analysis showed that *Af*Chi28 had a Glyco_hydro_18 domain (pfam02838, Met31-Asp353). Similarly, to most fungal chitinases, *Af*Chi28 lacks a chitin-binding domain (CBD) [25].

A BLAST search demonstrated that the sequences in the UniProtKB/Swiss-Prot da-tabase have 18.4–39.1% similarity to the sequence of *Af*Chi28. Thus, *Af*Chi28 is a newly discovered GH18 chitinase. Furthermore, the amino acid sequence of *Af*Chi28 was compared to 15 other different protein sequences of the GH 18 family with high homology from the UniProt database, as shown in Appendix A. A phylogenetic analysis of the GH18 family chitinases shows that they can be divided into three distinct subgroups previously identified by Ihrmark et al., [41]. Based on Appendix A, *Af*Chi28 had high homology with *Af*ChiB1 from *A. fumigatus*, *An*ChiB from *A. niger* CBS 513.88, *Cr*Chi1 from *Clonostachys rosea*, *Th*Chi42 from *Trichoderma harzianum*, *Sm*ChiB from *Serratia marcescens*, *Sm*ChiA from *S. marcescens*, *YERET*Chi1 from *Yersinia entomophaga* MH96, *Af*ChiB from *A. fumigatus*, and *YERET*Chi1 from *Y. entomophaga* MH96 in subgroup A of the GH18 family (Appendix A). Meanwhile, subgroup A corresponds to the Class V (fungal/bacterial) chitinases [42]. *Af*Chi28 belongs to the GH18 family subgroup A (Class V) chitinases.

Sequence analysis revealed that *Af*Chi28 contains the chitin-binding motif (SXGG) and a conserved catalytic motif (DxxDxDxE) commonly found in GH18 family members, in which the glutamate (E) residue is considered a proton donor in the catalytic process (Appendix A) [43].

### 2.3. Gene Coning of AfChi28

The open reading frame (ORF) of the full-length *Af*chi28 has 1116 bp, encoding 122 amino acids. Restriction enzymes *Nde*Ⅰ and *Hind* Ⅲ were used for enzymatic digestion of the expression plasmid DNA (Appendix A). The digested products were identified by agarose gel electrophoresis. After digestion, a 5304 bp vector band and a 1116 bp inserted gene fragment can be obtained, which are consistent with the theoretical values of 5304 bp and 1116 bp. This restriction digestion result showed that the recombinant plasmid was successfully constructed.

### 2.4. Expression and Purification of AfChi28

The chitinase gene, *Af*chi28, was bound to the vector pET-28a (+) and successfully expressed in *E**. coli* BL21 codon plus (DE3) RIL as an active protein. The recombinant *Af*Chi28 protein contained 122 amino acids and the expected molecular weight was 40.91 kDa. The recombinant *Af*Chi28 protein expresses at 30 °C with the addition of 0.1 mM isopropyl β-D-1-thiogalactopyranoside(IPTG). After IPTG induction and cell lysis, one major protein band was detected on the SDS-PAGE, with a molecular weight of approximately 40 kDa(Appendix A). The recombinant *Af*Chi28 showed a clear migration band on SDS-PAGE after adsorption and purification on Ni-NTA Bead 6FF column, and the molecular weight was about 40 kDa, which was consistent with its theoretical molecular weight (Appendix A, lane 6).

### 2.5. Enzyme Activity Assay of AfChi28 

In the enzyme reaction system, 800 µL of colloidal chitin at a concentration of 50 g/L was added with 200 µL 80 mg/mL of crude enzyme or 5 mg/mL of pure enzyme, reacted at 45 °C for 6 h and then extinguished in a boiling water bath. The enzymatic activity of the crude enzyme and pure enzyme was determined by Schales’ colorimetric method and the boiled crude enzyme or boiled pure enzyme were used as a blank control [44]. Crude extract of *Af*Chi28 hydrolyzed colloidal chitin with a specific activity of 0.737 mU/mg. *Af*Chi28 was purified to homogeneity by a Ni-NTA Affinity chromatography, with a specific activity of 52.414 mU/mg (for colloidal chitin), a purification factor of 71.12-fold and a recovery rate of 97.34% (Table 1).

### 2.6. Biochemical Characterization of AfChi28

The effects of pH and temperature on the activity of *Af*Chi28 were investigated. The effects of pH on the recombinant proteins were determined at pH 2.0–12.0 buffer systems. The crude enzyme solution of *Af*Chi28 has activity in the range of pH 2.0 to 12.0, and the optimum pH is 5.0 (Figure 2A). The pH is between 4.0–6.5, and the activity remains above 70%. The enzyme was stable in the pH range of 5.0–6.0, and the activity remained above 95% after pre-incubation at 4 °C for 24 h (Figure 2B). Similarly, the optimal pH of other chitinases from *Hydrogenophilus hirsch*ii KB-DZ44 [45], and *P. oxalicum* k10 [30] were within the range of 4.5–5.0. Therefore, *Af*Chi28 is a weak acid enzyme, giving it the potential for much wider application in industry.

To measure the optimum hydrolysis temperature, the recombinant protein was incubated with colloidal chitin as a substrate for 2 h at 25–80 °C. The crude enzyme solution of *Af*Chi28 is active over a wide temperature range, and the optimum temperature is 45 °C (Figure 2C). The enzyme is stable at 25–45 °C, after 90 min of incubation the activity remains greater than 90% (Figure 2D). Considering the thermal stability, the reaction temperature should be lower than 50 °C. In summary, in order to ensure the stability of *Af*Chi28, the reaction conditions were set to pH 5.0, 45 °C. Similarly, the optimal temperature of other chitinases from *P. oxalicum* k10 [30], *B. subtilis* [46], *Eisenia fetida* [11] *and A. fumigatus* CJ22-326 [12] were within a range of 40–45 °C.

As *Af*Chi28 maintains more than 90% of its activity when incubated for 90 min at 25 °C to 45 °C, its long-term stability at 30 °C and 45 °C was verified, in order to further investigate the thermal stability of the pure enzyme, its half-life and the residual power of the enzyme activity. As shown in Appendix A, *Af*Chi28 was largely degraded when incubated at 45 °C for 6 h, and basically degraded completely when incubated for 24 h. Micro-degradation occurred when incubated at 30 °C for 16 h. The results are consistent with those shown in Figure 2E, *Af*Chi2*8* maintained higher than 80% enzyme activity after incubation at 30 °C for 24 h, and reaches its half-life at about 60 h. However, *Af*Chi28 was completely inactivated after incubation at 45 °C for 16 h, and reached its half-life at about 7 h. In conclusion, the half-period of *Af*Chi28 incubated at 60 h at the incubation temperature of 30 °C was longer than 7 h at the incubation temperature of 45 °C. However, compared to chitinases from *A. fumigatus*YJ-407 [47] and *T. harzianum* GIM 3.442 [25], the thermal stability of *Af*Chi28 was much higher. Chitinases from *A. fumigatus* YJ-407 were incubated at 60 °C for 30 min and the residual enzyme activity was 20% of the original enzyme activity, which was less stable compared to *Af*Chi28, which was incubated at 60 °C for 2 h and still reached 20% of the original enzyme activity. Chitinase from *T. harzianum* GIM 3.442 was incubated at 45–50 °C, after 1 h incubation, the residual enzyme activity was 70% of the original enzyme activity. It is worth mentioning that *Af*Chi28 still reached 90% of the original enzyme activity after 2 h incubation at 45–50 °C.

### 2.7. Effect of Metal Ions and Chemical Reagents on Enzyme Activity

In a reaction mixture with 10 mM or 50 mM metal ions, estimation of the influence of metal ions was performed. These ions were Co^2+^, Na^+^, Mg^2+^, Cu^2+^, Fe^3+^, NH_4_^+^, Ca^2+^, Zn^2+^, Mn^2+^, K^+^, Ba^2+^, as well as EDTA, SDS, Tris and carbamide. The activity of the crude enzyme solution of *Af*Chi28 was set to 100%, when the compound concentration was 0 mM. As shown in Table 2, the addition of 10 or 50 mM of Na^+^ and K^+^ had significant positive effects on the *Af*Chi28 activity; especially when the concentration of Na+ was 10 mM or 50 mM, the enzyme activity was increased about 1.8-fold compared to the control. This result is consistent with the increased activity of most chitinases following the addition of K^+^ and Na^+^. For example, the activity of the wild chitinase from *C. tainanensis* CT01 [48], *P. oxalicum* [30] and *Vibrio harveyi* [49] was enhanced to 1.1, 1.8 and 1.4-fold, respectively, in the presence of K^+^. The activity of the wild chitinase from *C. tainanensis* CT01 [48] in the presence of Na^+^ was enhanced 1.01-fold. The concentrations of 50 mM of Co^2+^, Mg^2+^, NH_4_^+^, Ca^2+^ and Zn^2+^, slightly stimulated the activity of *Af*Chi28. In addition, Tris and carbamide also positively affected the activity of *Af*Chi28. Inhibition was observed in the presence of Cu^2+^, Fe^3+^, Mn^2+^, Ba^2+^, SDS or EDTA, especially Fe^3+^ or SDS. These findings suggest that the activity of *Af*Chi28 is affected by the metal ion species, which depend on disulfide bonds in the enzyme molecule [50]. These results provide a basis for the selection of metals or chemicals for industrial applications of *Af*Chi28.

### 2.8. The Substrate Specificity of AfChi28

To verify the ability of *Af*Chi28 to hydrolyze polysaccharides in addition to colloidal chitin, we performed substrate specificity experiments, as shown in the Table 3. *Af*Chi28 showed different advantages in the hydrolytic mode; *Af*Chi28 showed no significant activity when using CMC as a substrate. However, *Af*Chi28 had the highest specific activity when using colloidal chitin as a substrate. The specific activity of *Af*Chi28 is about 0.7134 mU/mg when it used colloidal chitin as a substrate, followed by 0.0237 mU/mg when using ball milled crab shell powder as a substrate (Table 3). In addition, *Af*Chi28 showed a similar activity of approximately 0.02 mU/mg when chitin powder and ball milled crab shell powder were used as substrates (Table 3), with a lower enzymatic activity than when colloidal chitin was used as a substrate. This was probably due to the insolubility of chitin powder and pretreated crab shell powder, and the high degree of polymerization of the substrate resulting in inaccessibility of the enzyme active center [1]. *Af*Chi28 also has a relatively low enzymatic activity, when the raw material was used as a substrate without deep treatment, offering the prospect of future developments to reduce contamination of acid and alkaline treated raw materials.

### 2.9. Kinetic Parameters 

The kinetic parameters of *Af*Chi28 were determined using colloidal chitin as the substrate for the hydrolysis reaction for 2 h. As shown in Appendix A, enzymes with lower *K_m_* have a higher affinity for the substrate, compared with the chitinase of P1724(∆cGH18) with the *K_m_* value of 2.1 ± 0.3 mg/mL [23], *Af*Chi28 had a slightly higher affinity than P1724(∆cGH18).

### 2.10. Hydrolysis Patterns of AfChi28 

Analysis of the hydrolysis products of the reaction of *Af*Chi28 with colloidal chitin at different reaction times was by thin layer chromatography. After the reaction mixture system was reacted at 45 °C for 0 h, 2 h, 4 h, 6 h, 8 h, 10 h, and 12 h respectively, 5 µL of the reaction product mixture concentrated by centrifugation, was analyzed by thin-layer chromatography (TLC). As shown in Appendix A, chitin oligosaccharides are produced within 0–24 h of the reaction of *Af*Chi28 with colloidal chitin, which includes GlcNAc, (GlcNAc)_2_, (GlcNAc)_3_ and (GlcNAc)_4_, indicating that *Af*Chi28 had endo-cleavage and exo-cleavage properties. GlcNAc is increasingly present after 2 h of reaction, as evidenced by the strip blots and (GlcNAc)_2_ and (GlcNAc)_3_ gradually increase between 4–8 h and decrease after 24 h in the reaction system. The shape of (GlcNAc)_4_ is relatively blurred in Appendix A.

The CAZy database shows that the GH18 family contains both endo-type chitinases, exo-type chitinases and a small amount of bifunctional chitinases. The endo-type chitinases cleave chitin to produce (GlcNAc)n (10 ≥ *n* ≥ 2), as in Chi46 from *Trichoderma harzianum,* GIM 3.442 [25] endo-cleaved colloidal chitin generates (GlcNAc)_2_, (GlcNAc)_3_ and (GlcNAc)_4_; exo-chitinases (β-glucosaminidase and chitobiosidase) can cleave chitin or (GlcNAc)n to generate GlcNAc or (GlcNAc)_2_, as in chitinase from *Paenibacillus*
*chitinolyticus* strain UMBR 0002 [51] exo-cleaved colloidal chitin generates (GlcNAc)_2_. However, in the existing research reports, there are very few bifunctional chitinases with endo-cleavage and exo-cleavage functions. For example, chitinase from *P. oxalicum* k10 [30] was capable of hydrolyzing chitin to (GlcNAc)_3_, (GlcNAc)_2_ and GlcNAc. Chitinase Chit42 from *Trichoderma harzianum* [52] was capable of hydrolyzing chitin to (GlcNAc)_4_, (GlcNAc)_3_, (GlcNAc)_2_ and GlcNAc. The results of this study is similar to the above two bifunctional chitinases, which exhibit endo-cleavage and exo-cleavage activity and degrade colloidal chitin to produce (GlcNAc)_4_, (GlcNAc)_3_, (GlcNAc)_2_ and GlcNAc. From the perspective of catalytic mechanism, broad-specificity chitinase is more efficient than single-activity chitinase, and is a good candidate for green transformation of chitin waste, which has great value in research and application [30].

### 2.11. HPLC and Q Exactive LC-MS Analysis of Hydrolysates Produced by AfChi28

The colloidal chitin was hydrolyzed by the enzyme *Af*Chi28 to make it more mobile and most of the insoluble chitin was converted to soluble oligomers. HPLC spectra of the hydrolyzed products after 2 h, 4 h, 6 h, 8 h, 10 h, 12 h and 24 h reactions, respectively, are shown in Figure 3. At a hydrolysis reaction time of 2 h, one major peak and three minor peaks were detected corresponding to GlcNAc, (GlcNAc)_2_, (GlcNAc)_3_ and (GlcNAc)_4_ on the standards, respectively, at the same peak times. However, at a hydrolysis time of 12 h, (GlcNAc)_4_ was completely hydrolyzed. At a hydrolysis reaction time of 12 h, the total amount of (GlcNAc)_2_ and (GlcNAc)_3_ increased by approximately 50% compared to a hydrolysis reaction time of 6 h. The results showed echoes of the experimental results from TLC (Appendix A). These data indicate successful enzymatic hydrolysis of chitin oligomers from colloidal chitin.

As show in Figure 3 and Appendix A, chitin oligosaccharides with a higher degree of polymerization can be obtained when the reaction time is 6 h, but the concentration of chitin oligosaccharide of DP_4_ is sightly low. In order to further verify the main product of colloidal chitin *Af*Chi28 hydrolysis reaction time of 6 h, Q exactive LC-MS was used to analyze the products (Appendix A). The hydrolysis products were analyzed with reference to the molecular weight size of the chitin oligosaccharides [51]. As can be seen from the characteristic peaks of the products corresponding to the molecular weights, the mixture of reaction products with a hydrolysis time of 6 h contains GlcNAc, (GlcNAc)_2_, (GlcNAc)_3_, (GlcNAc)_4_. Thus, the analysis shows that *Af*Chi28 has endo-cleavage and exo-cleavage functions. However, the exact catalytic mechanism of *Af*Chi28 hydrolysis of chitin needs to be further investigated.

### 2.12. Structural Analysis of AfChi28 and Molecular Docking Simulation

The structure of *Af*Chi28, as deduced by AlphaFold2 calculations [53], shows a structural feature previously described for other GH18 chitinases, namely a (β/α)_8_ TIM barrel fold with an additional α/β structural domain, which is composed of five antiparallel β-strands flanked by one α-helix, inserted in the loop connecting helix α8 and barrel chain β8 (Figure 4A) [52,54]. This extra domain might help to provide the groove shape for the active site. The catalytic signature motif of GH18 chitinases, DXDXE (residues150–154), is located between β4-α4 (Figure 4). The overall structure of *Af*Chi28 is similar to that of several other fungal GH18 chitinases, with a root mean square derivation (rmsd) of 1.6 Å with *An*ChiB from *Aspergillus niger* (395 Cα atoms) [55], 1.4 Å with *Cr*Chi1 from *Clonostachys rosea* (391 Cα atoms) [56], and 1.5 Å with *Af*ChiB1 from *A. fumigatus* (339 Cα atoms) [57]. Meanwhile, some conformational differences exist in the connecting loops, many of which harbor insertions or deletions. Compared with the major secreted chitinase of *A. fumigatus*, *Af*ChiB1, *Af*Chi28 didn’t contain a signal peptide. In the active-site cleft of *Af*Chi28, Gly113 replaces a bulky Trp137 in *Af*ChiB1 and causes the active-site cleft of *Af*Chi28 to be larger than in *Af*ChiB1. It is clearly demonstrated that *Af*Chi28 can catalyze the bulkier substrate.

Substrates within the active site channel were simulated by docking (GlcNAc)_6_ with the AlphaFold2 model of *Af*Chi28 (Figure 4B). It was found to form a deep groove on the surface of the protein. Details of the proposed interaction with oligosaccharides are shown in Figure 4C. The structure of (GlcNAc)_6_ binding of *Af*Chi28 exhibits some distorted differences, particularly at the negative subsites, suggesting a conformational change in the enzyme upon substrate binding. Molecular docking revealed that the substrate binding cavity of *Af*Chi28 contains six glycosidic binding sites, ranging from −4 to +2, and that the predicted catalytic residue of *Af*Chi28, Glu154, is spatially close to the glycosidic bond between −1 and +1. The amino acid residues on the substrate binding cavity of *Af*Chi28 stabilize the substrate by forming hydrogen bonds with (GlcNAc)_6_. His155 stabilizes the boat conformation of GlcNAc at the +1 position through hydrogen bonding, and this conformation is essential for initiating the enzymatic reaction. Glu154, Trp39 and Asn38 form hydrogen bonds with the O6 hydroxyl group of the sugar group at the −4, −3 and −1 positions respectively. In contrast, the acetylamino group at the C2 position of the glycosyl group at the −1 site forms a hydrogen bond interaction with the side chain of Arg269.

## 3. Materials and Methods

### 3.1. Strains and Materials 

The *Af*Chi28(1G02800 class Ⅴ chitinase, Gene Bank: OK302920) gene was derived from *Aspergillus fumigatus* df347. Expression vector pET28a (+) and *A. fumigatus* df347 are kept in our laboratory. The pET28a (+) vector was used for the cloning and expression of the target gene. *E. coli* DH5a and *E. coli* BL21 codon plus (DE3) RIL were used for gene cloning and heterologous expression, respectively. 2X Taq Master Mix (Dye Plus) (Vazyme Biotech Co., Ltd, Nanjing, China), 2XPhanta^®^ Max Master Mix (Dye Plus) (Vazyme Biotech Co., Ltd, Nanjing, China), ClonExpress^®^ Ultra One Step Cloning Kit (Vazyme Biotech Co., Ltd, Nanjing, China) and the restriction enzymes were purchased from Vazyme (Nanjing, China). Chitin powder (from shrimp shells) was purchased from Aladdin (Shanghai, China). Chitin oligosaccharides as a standard was purchased from Tokyo Chemical Industry Co., Ltd (Tokyo, Japan). Ni-NTA Beads 6FF column used for affinity chromatography was purchased from Smart lifesciences (Changzhou, China).

### 3.2. Sequence Analysis 

CAZy (http://www.cazy.org/Home.html, accessed on 3 May 2021)) was used to screen potential target genes of the GH family 18. BLASTN was used to compare the homology of the target genome to sequences in the NCBI database (https://blast.ncbi.nlm.nih.gov, accessed on 25 May 2021). SignalP 5.0 server (http://www.cbs.dtu.dk/services/SignalP/, accessed on 1 June 2021) was used to predict the signal peptide of the protein sequence. Smart (http://smart.embl.de/, accessed on 1 June 2021) was used to predict structural domains. ExPASy (https://web.expasy.org/translate/, accessed on 1 June 2021) was used to translate nucleotide sequences into amino acid sequences. ExPASy (http://www.expasy.org/tool, accessed on 1 June 2021) was used to predict protein molecular weight (MW) and Isoelectric point (pI) [58]. Snapgene was used for the rapid construction of plasmid maps (Version 5.2, www.snapgene.com, accessed on 27 May 2021). MEGA-X generated evolutionary trees through the neighbor joining method (Version 11, Home Page, China) [59]. Amino acid sequence alignments were performed on the ClustalX [43] server, using default settings, and sequences were optimized in the ESPript 3.0 online tool (https://espript.ibcp.fr/ESPript/ESPript/, accessed on 8 April 2022).

### 3.3. Isolation of Strains with Chitinolytic Activity

Sediment samples were collected in Mangrove Reserve, Qinzhou, Guangxi Autonomous Region, China (21°86′01.95″ N 108°60′68.54″ E). after enriching the soil in LB medium for 2–3 days, the culture was then diluted to 10^−8^ with sterile water, then spread on separation medium (The separation medium was formulated with 0.5% colloidal chitin, 0.05% NaCl, 0.05% MgSO_4_, 0.075% KH_2_PO_4_, 0.01% FeSO_4_, 0.3% (NH_4_)_2_SO_4_, 2% agar power and adjusted to pH 6.5.) with chitin as the sole carbon source. Individual colonies of screened fungi were cultured individually in PDB medium and preserved in glycerol, namely df347.

### 3.4. Strain Identification

Genomic DNA was extracted from the screened strains using a fungal DNA kit (Sangon Biotech, Shanghai, China). To identify the strain, its ITS rDNA gene sequence was amplified using primers (ITS1: 5′TCCGTAGGTGAACCTGCGG3’; ITS4: 5′TCCTCCGCTTATTGATATGC3’), and then sequenced by Sangon (Sangon Biotech, Shanghai, China). The gene sequences were compared with Gene Bank from NCBI database using the BLAST program to calculate sequence homology. Phylogenetic trees were generated using MEGA-X software with analysis of the Neighbor-Joining Algorithm.

### 3.5. Preparation of cDNA 

TRIzol reagent was used to extract the total RNA of *Aspergillus fumigatus* df347, as described by Chomczynski, with slight modification [60]. *A. fumigatus* df347 was fermented in Potato Dextrose Broth (PDB) medium at 28 °C for 16 hours. The mycelium was freeze-dried with liquid nitrogen in stages, and then the freeze-dried mycelium was ground in a precooled mortar to make it into powder. The ground powder was transferred to an eppendorf tube where 1 mL Triquick Reagent was added (powder: Triquick Reagent = 0.1 g:1 mL). The mixture was keep at room temperature (RT) for 10 min to mix well, then centrifuge at 12,744× *g* for 15 min at 4 °C. 200 µL of chloroform was added to the supernatant. After vortexing for 15 s, the mixture was kept at RT for 15 min, followed by centrifugation at 12,744× *g* at 4 °C for 15 min. The supernatant was transferred to a new centrifuge tube, but not precipitate. Then, 400 µL isopropanol was added to the supernatant, incubated at room temperature (RT) for 10 min, and centrifuged at 12,744× *g* at 4 °C for 10 min. 1 mL of pre-chilled 75% ethanol was added to the pellet and incubated for 2 min, then the mixture was centrifuged at 12,744× *g* for 5 min at 4 °C. The supernatant was discarded completely, and the RNA pellet was dried at RT for 10 min. Finally, the RNA pellet was dissolved in 40 µL 0.1% DEPC water at RT. The RNA was reverse transcribed to cDNA using the Prime Script II 1st Strand cDNA (TaKaRa) kit (TaKaRa BIO INC, Japan). RNA and cDNA were all stored at −80 °C before further use.

### 3.6. Gene Cloning of AfChi28 

For heterologous expression of *Af*Chi28 (Gene Bank: OK302920) in *E. coli*, the target gene was amplified from *Aspergillus fumigatus* df347 cDNA. Restriction endonucleases *NdeⅠ* and *HindⅢ* sites were added to the forward and reverse primers (F:5′GTGCCGCGCGGCAGCCATATGATGTTCTCCGGATCCATCTTCC3′; R:5′CTCGAGTGCGGCCGCAAGCTTTCACATATCATGCAAGGTCTTATACCC3′), respectively. The PCR product was gel-purified, digested with *NdeⅠ* and *HindⅢ*, and cloned into the corresponding sites of the pET28a (+) vector. The recombinant plasmid was transformed into competent cells of *Escherichia coli* DH5a. The positive *E. coli* DH5α recombinants were identified by PCR and DNA sequencing (Sangon Biotech, Shanghai, China).

### 3.7. Construction of Expression Vector in E. coli BL21 Codon plus (DE3) RIL

A positive *E. coli* DH5α transformation was screened and cultivated in Luria–Bertani medium (LB medium) with 0.1 mM kanamycin (Kan) for obtaining multicopy recombinant plasmids. Subsequently, the recombinant plasmid was extracted from the expanded DH5a by a GeneJET Plasmid Minipreo Kit (thermo scientific), and transformed into *E. coli* BL21 codon plus (DE3) RIL competent cells for expression. The positive multicopy transformation was incubated in LB medium at 37 °C until the OD_600_ reached 0.6–0.8. And then 0.1 mM IPTG was added to the mixture at 30 °C for 8 h for expression. Subsequently, the fermentation broth was centrifuged at 8,684× *g* for 20 min at 4 °C, and the resulting precipitate was collected. The precipitate was resuspended in 15 mL of phosphate buffer solution (pH = 6.0), and the mixture was disrupted by sonication on ice (setting: amplitude 1 × 10^−6^ w, ultrasonic time 1 s, pause time 1 s, total time 60 min), followed by centrifugation at 13,593× *g* at 4 °C for 30 min. After high-speed centrifugation, the target proteins were identified by SDS-PAGE electrophoresis using the cell lysate and supernatant.

### 3.8. Purification of Recombinant AfChi28

The supernatant was applied to the Ni-NTA Beads 6FF column immediately and strictly controlled at a temperature of 4 °C. The column was washed with 1X binding buffer (50 mM Tris, 300 mM NaCl, pH 7.5, and 10% glycerin). The crude enzyme was loaded onto the Ni-NTA Beads 6FF column (Smart lifesciences, Changzhou, China). Subsequently, the column was washed thoroughly with phosphate buffer (50 mM Tris, 300 mM NaCl, pH 7.5, 10% glycerin, with 50 mM imidazole) to remove the unbound proteins. The target protein bound to the column was eluted with elution buffer (50 Mm Tris, 300 mM NaCl, pH 7.5, and 10% glycerin, with 500 mM Imidazole). After elution, *Af*Chi28 was desalted by a HiTrap TM Desalting column (GE Healthcare, Beijing, China). The desalting column was washed with desalting buffer (20 Mm Tris, 100 Mm NaCl, pH 7.5, and 10% glycerin) to balance the column. When the OD_420_ reading of the UV detector stabilized, all the purified enzyme was loaded onto the HiTrap TM Desalting column, buffer exchange was done with the same buffer at 1 mL/min. To collect the desalted solution it was loaded onto centrifugal filter units (30,000) (Merck Millipore Ltd., Kenilworth, NJ, USA) for concentration. The recombinant protein was then stored at −20 °C for further use. The molecular weight and purity of *Af*Chi28 was verified by SDS-PAGE, which was performed according to the method of Laemmli using 12.5% polyacrylamide gels (Gold-tech Biotechnology Co., Ltd., Nanning, China) containing 0.1% SDS [61] (Solarbio Science & Technology Co., Ltd., Beijing, China) [61]. The protein was stained with Coomassie blue dye. Protein ladder (15−150 kDa) was purchased from Vazyme (Nanjing, China). The protein samples were denatured by heating for 10 min at 99 °C before running the gel electrophoresis.

### 3.9. Enzyme Activity Assay of AfChi28 

Chitinase activity was assayed by the method described previously [27,44] with minor modifications. The activities of purified *Af*Chi28 with colloidal chitin as substrates were assayed by Schales’ procedure [44]. Chitin powder was mixed with concentrated HCl at a ratio of 1:12 (g/mL), stirred until a paste was formed and left at 4 °C for 24 h. Subsequently, 2 L of pre-cooled 95% ethanol was added, stirred vigorously, and left overnight to precipitate the colloidal chitin. The precipitate was collected by centrifugation at 12,744× *g* for 20 min at 4 °C. After high-speed centrifugation, to collect colloidal chitin, precipitate was washed repeatedly with ddH_2_O to pH 5.0, and the volume was adjusted to 100 mL with 0.2 M citric acid-disodium hydrogen phosphate buffer solution (pH 5.0), which was 5% colloidal chitin. The reaction system consisted of 800 µL 5% substrate solution (*w*/*v*, pH 5.0) and 200 µL 80 mg/mL enzyme solution. The control system consisted of 800 µL 5% substrate solution (*w*/*v*, pH 5.0) and 200 µL 0.2 M disodium hydrogen phosphate–sodium dihydrogen phosphate buffer solution (pH = 5.0). The system was kept at 45 °C for 1 h, after which the reaction was stopped by adding 1 mL Schales’ reagent and subsequently boiled in a water bath for 10 min. The reaction system was cooled in an ice-water bath and centrifuged at 12,744× *g* for 20 min at 4 °C; the supernatant was collected, and the absorbance was measured at 420 nm. One unit of enzymatic activity is defined as the amount of *Af*Chi28 required to release 1 µmol of *N*-acetylglucosamine per minute under the assay conditions. *N*-acetylglucosamine were used as standards, respectively.

### 3.10. Biochemical Characteristics of AfChi28 

The optimal pH of *Af*Chi28 was determined by measuring the enzyme activity in 100 mM of various pH buffers at 45 °C. The buffers were citric acid-disodium hydrogen phosphate buffer solution (pH 2.0–8.0), glycine–NaOH buffer (pH 8.0–12.0). To determine the pH stability of chitinase activity, the above-described recombinant protein with a different buffer pH was incubated for 24 h at 4 °C. The residual enzyme activity in the reaction mixture was measured under the standard enzyme reaction mixture and assay condition.

The optimal temperature of *Af*Chi28 was determined in 100 mM citrate buffer (pH 5.0) at different temperatures (25–80 °C). In order to determine the thermal stability of *Af*Chi28, the residual activity of the enzyme was measured after incubation in 100 mM citrate buffer pH 5.0 at 25–80 °C for 90 min. And then the remaining enzyme activity was measured in the standard enzyme reaction mixture and assay condition.

To verify the stability of pure enzyme at temperatures of 30 °C and 45 °C, *Af*Chi28 purified by Ni-NTA Beads 6FF column was incubated in water baths at different temperatures (30 °C and 45 °C). Samples were taken at different incubation times (2 h, 4 h, 6 h, 8 h, 12 h, 16 h, 24 h, 36 h and 48 h). The sampled proteins at different induction temperatures were analyzed by SDS-PAGE. And then the remaining enzyme activity was measured in the standard enzyme reaction mixture and assay condition.

### 3.11. Effect of Metal Ions and Various Chemicals on Enzyme Activity

The effects of metal ions (i.e., Fe^3+^, NH_4_^+^, Ca^2+^, Zn^2+^, Mn^2+^, K^+^ and Ba^2+^) and chemicals (i.e., Tris-(hydroxymethyl)-aminomethane (Tris), SDS, ethylenediaminetetraacetic acid (EDTA), Urea) on the activities of *Af*Chi28 were determined at the concentrations of 5 and 10 mM. These compounds were added to the standard enzyme reaction mixture accordingly, and then tested under standard assay conditions.

All data were analyzed using the software GraphPad Prism 8.0 (GraphPad Software, La Jolla, CA, USA).

### 3.12. Substrate Specificity and Kinetic Parameters

The substrate specificity of *Af*Chi28 was determined by measuring its specific activity using different substrates in 100 mM citrate buffer (pH 5.0) at 45 °C. Substrates included colloidal chitin, chitin powder, carboxymethyl cellulose (CMC), chitosan powder, and pretreated crab shell powder. After the reaction between the polysaccharide substrate and *Af*Chi28, 1 mL Schales’ reagent was added to the system mixture and the absorbance was measured at 420 nm.

The kinetic parameters of *Af*Chi28 on colloidal chitin were determined. Recombinant proteins were reacted with different concentrations (0.0004–0.02 g/mL) of colloidal chitin in 100 mM citric acid–disodium hydrogen phosphate buffer solution (pH 5.0) in a water bath at 45 °C for 2 h. *Km* values and *K_cat_* values were calculated separately from the kinetic data using Origin 2021 software.

### 3.13. TLC Analysis of Colloidal Chitin Hydrolysates Produced by AfChi28

The identification method of the enzymatic hydrolysate was as described previously [62], with minor modifications. The standard enzyme reaction mixture was measured under standard conditions, and stopped by boiling for 10 min, followed by centrifugation at 12,744× *g* at 4 °C for 10 min. After centrifugation, the supernatant of the mixture was concentrated about 20 times with a vacuum centrifuge, and then spotted on a Silicagel 60-F254 aluminum sheet. The TLC plate was developed in a solvent system of N-butanol: acetic acid: H_2_O (2:1:1, *v*/*v v*/*v*). The developed spots were analyzed by spraying the aniline–diphenylamine reagent (4 mL of aniline, 4 g of diphenylamine, 200 mL of acetone, and 30 mL of 85% phosphoric acid), and then heating it in a hot air oven set at 80 °C for 20 min. The hydrolysates were estimated using a chitin oligosaccharides standard.

### 3.14. HPLC and Q Exactive LC-MS Analysis of Colloidal Chitin Hydrolysates Produced by AfChi28

Colloidal chitin undergoes enzymatic reaction under standard conditions via chitinase. The reaction mixture was inactivated in a water bath for 10 min and centrifuged at 12,744× *g* at 4 °C for 30 min. After centrifugation, the supernatant was filtered through a 0.22 µm nylon filter membrane and analyzed by HPLC. For *N*-acetyl COSs, the amount of released reducing sugars was determined by HPLC. For HPLC analysis, the hydrolysis samples were subjected to an HPLC system (UltiMate 3000 series) equipped with a Luna^®^ 5 μm NH_2_ column (4.6 mm × 250 mm, Phenomenex Inc., Torrance, CA, USA) and a refractive index detector (RID). 70% acetonitrile solution (*v*/*v*) was used as mobile phase with a flow rate of 0.6 mL/min. The column oven was kept at room temperature.

After the enzyme reaction, the enzymatic hydrolysis product in the supernatant were concentrated about 20 times with a vacuum centrifuge. The supernatant was filtered through a 0.22 µm nylon filter membrane and analyzed by Q Exactive LC-MS. Afterward, the samples were analyzed with Q Exactive LC-MS coupled with an HPLC system using Hypersil GOLD C18. The solvents used were as follows: (A) 0.1% formic acid in H_2_O and (D) CH_3_CN. The LC-MS (product ion scan) had a low rate of 0.4 mL/min in gradient mode and gradient elution was performed at room temperature by the following method: 5% D in A with a linear gradient for 2 min, followed by 5% to 95% D in A with gradient for 8 min, remaining at 95% D in A for an additional 7 min and then an immediate return to the initial conditions for a 5 min re equilibration period as a cycle.

### 3.15. Structure Prediction and Molecular Docking

AlphaFold2 was used to predict the three-dimensional structure of *Af*Chi28. Ligand molecule (GlcNAc)_6_ was drawn by ChemDraw software. The AutoDock Vina and flexible docking methods were used to separately dock *Af*Chi28. The Asp152 of AfChi28 was used as the center of the box respectively. PyMOL software was used for structure visualization and detailed interaction analysis.

## 4. Conclusions

Chitinases are an important way of converting chitin into bio-functional materials and there is a lack of high performance, low cost chitinases. In this study, a chitinase *Af*chi28 was identified from a marine *A. fumigatus* df347 and characterized. *Af*Chi28 is an acido-halotolerant and temperature-resistant bifunctional enzyme with an enzymatic activity of 52.414 mU/mg. Among the hydrolysis products, *Af*Chi28 efficiently hydrolyses chitin to produce highly aggregated chitin oligosaccharides, of which the hydrolysis products contain GlcNAc, (GlcNAc)_2_, (GlcNAc)_3_ and (GlcNAc)_4_. These properties made this enzyme a promising candidate for the green industrial conversion of bulk marine chitin wastes into chitin oligosaccharides with high added value. The 3D structure of *Af*Chi28 was predicted by AlphaFold2 to demonstrate that it has the motif (DxxDxDxE) at the end of strand β5, with E154 as the catalytic residue in the middle of the open of the TIM barrel like other Family 18 chitinases. A (GlcNAc)_6_ molecule was shown to bind to *Af*Chi28 in subsites −4 to +2 in the substrate-binding domain which was further verified by a molecular docking study. Our findings provide a certain research contribution for elucidating the catalytic mechanism of chitinases from marine fungi.

## Figures and Tables

**Figure 1 marinedrugs-20-00520-f001:**
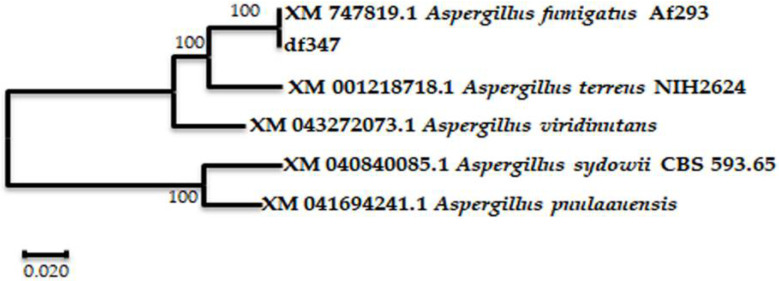
Phylogenetic tree of strain df347 based on the homology of ITS rDNA sequences. Tree scale: 0.020 substitutions per nucleotide position.

**Figure 2 marinedrugs-20-00520-f002:**
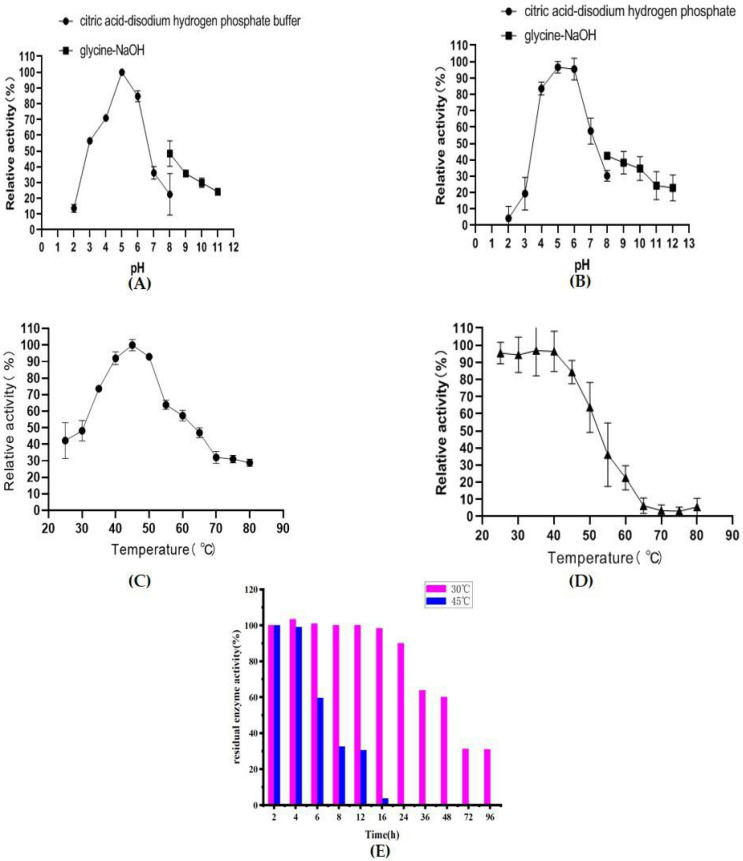
Effects of temperature and pH on the activity of *Af*Chi28. (**A**) Optimum temperature. (**B**) Temperature stability. (**C**) Optimum pH. (**D**) pH stability. (**E**) Temperature Stability at 30 °C and 45 °C. All data are expressed as the mean ± standard deviation (S.D.) of three experiments.

**Figure 3 marinedrugs-20-00520-f003:**
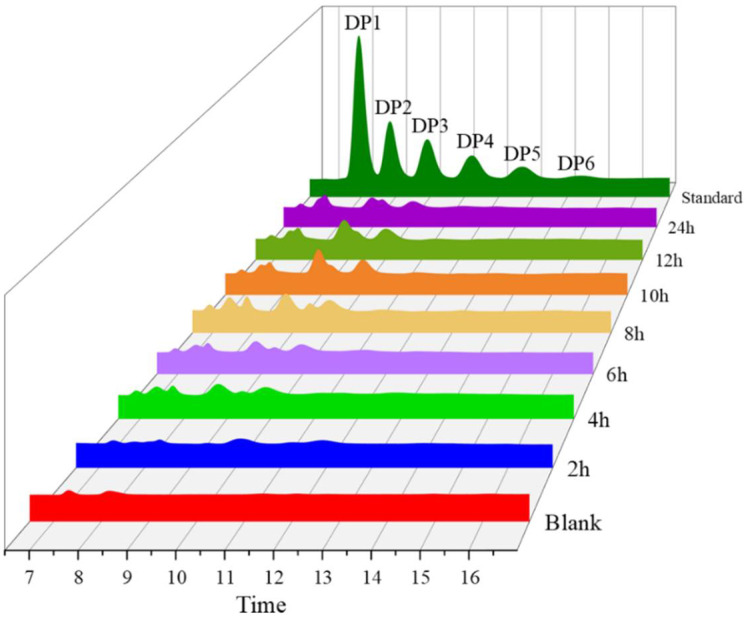
HPLC analysis of the hydrolysis products of colloidal chitin by *Af*Chi28. The *Af*Chi28 was incubated with 50 g/L of the colloidal chitin in the reaction mixture for different time periods at 45 °C. The reaction products were analyzed by HPLC. The top profile shows the standard mixture of chitin oligosaccharides from DP1 to DP6.

**Figure 4 marinedrugs-20-00520-f004:**
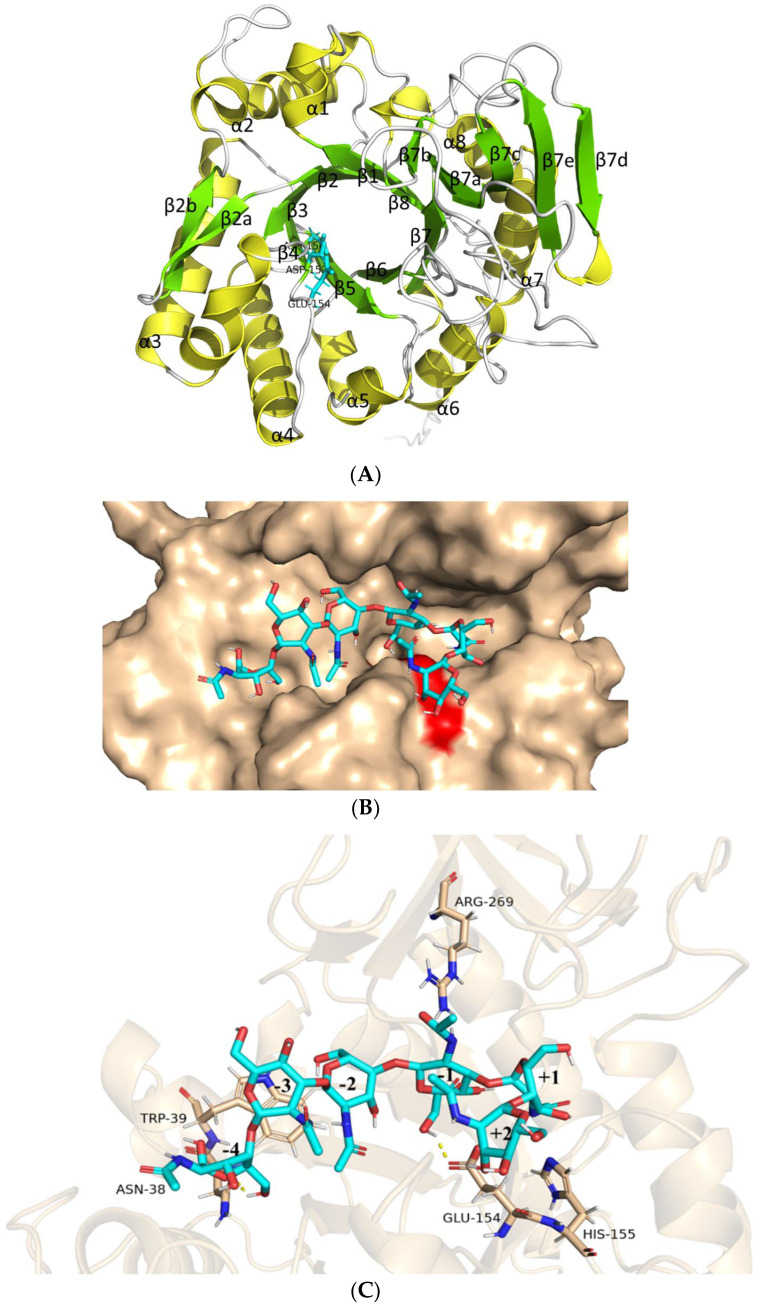
The active site of *Af*Chi28. (**A**) Crystal structure of *Af*Chi28. The *Af*Chi28 structure is represented as a cartoon. The α-helices, β-strands, and loops are shown in tv_yellow, chartreuse, and white, respectively. The catalytic residues (Asp ^150^, Asp ^152^, and Glu ^154^) are shown as cyan sticks. (**B**) (GlcNAc)_6_ mimics the active site by molecularly docking with the AlphaFold2 putative protein structure of *Af*Chi28. the molecular surface of *Af*Chi28 shows blue sugar sticks. Catalytic Asp152 and Glu154 are red. (**C**) Details of the atomic interactions between (GlcNAc)_6_ and *Af*Chi28-related residues are indicated by sticks. Highlighted catalytic residues and proposed hydrogen bonds are indicated by dashed lines.

**Table 1 marinedrugs-20-00520-t001:** Purification summary of *Af*Chi28.

Purification Step	Total Activity (mU)	Total Protein (mg)	Specific Activity (mU/mg)	Purification Factor (-Fold)	Recovery Yield (%)
Crude extract	577.248	783.24	0.737	1.0	100
purified protein eluate	561.878	10.72	52.414	71.12	97.34
Gel filtration chromatography	156.837	0.242	648.088	879.36	27.17

**Table 2 marinedrugs-20-00520-t002:** Effect of different metal ions and chemical reagents on *Af*Chi28 ^a^.

Additives	Relative Activity (%)	
	0.01 M	0.05 M
Control	100.00	100.00
Cobaltous (Co^2+^)	75.40	131.35
Sodium (Na^+^)	181.79	173.34
Magnesium (Mg^2+^)	53.50	145.73
Copper (Cu^2+^)	24.53	62.19
Iron (Fe^3+^)	15.30	62.61
Ammonium(NH_4_^+^)	83.51	122.90
Calcium (Ca^2+^)	92.29	160.70
Zinc (Zn^2+^)	34.44	134.76
Manganese(Mn^2+^)	47.80	94.28
Potassium (K^+^ )	111.03	147.61
Barium(Ba^2+^)	40.18	73.21
Tris	105.75	155.56
SDS	0	0
EDTA	52.56	92.34
Carbamide	106.49	120.77

^a^ Values represent means (*n* = 3) relative to the untreated control samples.

**Table 3 marinedrugs-20-00520-t003:** Substrate specificity of *Af*Chi28.

Substrate	Specific Activity (mU/mg) ^a^
Colloidal chitin	0.7134
Chitin powder	0.0137
CMC	None
Chitosan powder	None
Ball milled crab shell powder	0.0237

^a^ Specific activity of *Af*Chi28 was determined by measuring the enzyme activity in 100 mM citrate buffer (pH 5.0) at 45 °C using 5% (*w*/*v*) of various substrates. The data is the mean ± standard deviation (S.D.) of three experiments.

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
