# Peer review of "The Discovery, Enzymatic Characterization and Functional Analysis of a Newly Isolated Chitinase from Marine-Derived Fungus Aspergillus fumigatus df347"

_marinedrugs, 2022, doi:10.3390/md20080520_

Round 1

Reviewer 1 Report

Title: The Discovery, Enzymatic Characterization and Functional 2 Analysis of a Novel Chitinase from Marine-Derived Fungus 3 Aspergillus fumigatus df347

Authors: Wu Y. et al.

Specific comments

1. Figure 1. Please include a evolutionary scale bar inside the tree figure and legend to present the evolutionary distance among the strains.

2. Figure 4. This figure can be deleted or if necessary, moved to supplementary file.

3. The more discussion on differences b/w AFchi28 and the previously reported chitinases can strengthen this manuscript.

Reviewer 2 Report

Review to the manuscript entitled “The Discovery, Enzymatic Characterization and Functional Analysis of a Novel Chitinase from Marine-Derived Fungus Aspergillus fumigatus df347” by Ya-li Wu et al. submitted to Marine Drugs.

The manuscript of a research paper focuses on biochemical characterization of chitinase from Aspergillus fumigatus df347 isolated from marine setting. The study has been thoroughly conducted with characterization of properties of AfChi28, structure prediction and evaluation of the released products. As chitin is an abundant side-product of seafood production, e.g. from crustaceans, the valorisation of chitin effectively into functional oligosaccharides is of high importance for the industry. The aims and scope of the manuscript is in accordance with that of the journal.    

General remarks:

The manuscript is generally adequately structured and composed, and clearly presented but some additions or corrections have to be made before the manuscript can be considered for publication. The manuscript is written in an English language on academic style but some corrections in wording and grammar might be needed to enhance clarity. The use of professional English text editing is encouraged. The compliance of the text editing to the recommendations to the journal should be checked.  

According to CAZy database many EC 3.2.1.14 chitinases from Aspergillus sp. have already been characterized and in PDB database the 3D structure of Aspergillus fumigatus chitinases are available. This has not been discussed in the manuscript. GH18 family chitinases from various organisms have been extensively studied. Therefore, the novelty aspects of the study have to be highlighted more clearly.

The Introduction could provide more comprehensive overview of the chitin degradation pathways. Citinases are introduced but recently discovered lytic polysaccharide monooxygenases (LPMOs) that also act efficiently on crystal chitin are not mentioned. In the end of the manuscript (line 571), LPMOs are mentioned without giving the explanation of the abbreviation or the vital context.

The Results section lacks the summary of information regarding the isolated Aspergillus strain and identification of chitinolytic activity. This is described in the Materials and Methods section but no results have been shown on it. Is the fungal strain deposited to the microbial collection?

The Discussion part is blended to the Results and it is rather minimal and should be improved.

Specific comments:

Title, abstract, results (lines 3, 20, 116). The use of word “novel” should be justified and made clearer. It has to be clarified whether AfChi28 truly a novel enzyme or rather newly isolated chitinase. Which features are suggesting that the chitinase enzyme is novel? Members of GH18 family, also harbouring AfChi28, are widely recognised as classical chitinases. Also, based on PDB and CAZy databases the A. fumigatus chitinases ChiA1 and ChiB1 have been characterized in detail and their 3D structures are available.   

Throughout the text. In “N-acetyl” the N should be written in italics. Some abbreviations are explained twice - DP, GlcNAc.

Line 50. Abbreviation “COS” should be explained.

Line 66. The term “plant exfoliating fences” should be checked.

Lines 73-77. The relevant references should be provided.

Line 80. The salinity of the isolation location should be provided.

Lines 103-104. It has to be discussed how the secretion signal-negative enzyme can stay inside of the cell whereas the substrate (chitin) is located extracellularly.

Fig. 2. It has to be noted that Neosartorya fumigate is synonym for Aspergillus fumigatus to avoid confusion.

Fig. 4. It is technical detail and should be presented in Supplementary information.

Fig. 6. The information on the panels and the legend of the figure are not concordant.

The references to the figures in the text should be carefully checked as they do not seem to be constantly accurate.

Lines 297-298. How the optimum time point was verified? No quantitative data are shown on chitin oligosaccharide formation.

Line 376. Was the sample soli or sediment, as mentioned earlier?

Reviewer 3 Report

The paper needs extensive revision.

Abstract must be rephrase: in particular the first two sentences.

Introduction, for what I know, chitin is present in all fungi, not only pathogens. 

lane 82: why The chitinase gene (AfChi28) was mined from the rDNA?

Results

- line 97: why teorethical promers?

- line 128: this phrase is a repetiton.

- was the sequence deposited in the data bank? I could not find any acc. number.

- in the sequence alignment and phylogenetic tree, the acc nos of all sequences used must be reported.

- Why did they test the effects of different Metal ions? How they can conclude that this new chitinase is more efficient?

Material and methods

- this section lacks the primers used for ITS amplification, Furthermore, it lacks teh description of the method used for teh construction of the phylogenetic tree. 

Conclusion

-lane 564: witout mutagenesis tests, I would be more mild in this statement, simply saying that it is a contribution.

Reviewer 4 Report

The manuscript entitled “The Discovery, Enzymatic Characterization and Functional Analysis of a Novel Chitinase from Marine-Derived Fungus Aspergillus fumigatus df347” demonstrates the functional properties and catalytic mechanism of chitinases from marine fungi. The manuscript is interesting and could be proceeded further for publication in Marine Drugs after a major revision of the comments below.

The introduction is poorly written. It could be improved through the collection of updated reports on chitin and Chitinase from public literature.

Lines 39-41: the authors mentioned “Chitin, linked by repeated β-1, 4-N-acetylglucosamine (GlcNAc) units, is the second most abundant polysaccharide after cellulose. They are mainly found in the cell walls of fungal pathogens and exoskeleton material of arthropods and crustaceans[1,2]”. This info is misleading. Chitin has also been reported in the cell walls of coralline algae e.g., Sci Rep 4, 6162 (2014). https://doi.org/10.1038/srep06162. As mentioned above, the authors should include background/intro with the latest about Chitinases as well.

Fig. 7: the results of this figure are vague. It is not clear what experiment has been used for that. Is it SDS-PAGE or Blotting or any other data? The figure caption should be written clearly which could help the readers to understand the results.

Materials and methods: SDS-PAGE details including the manufacturer’s name should be included in this section.  

There is no discussion in this manuscript. A cutting-edge discussion is required to validate of authors’ current findings.

The manuscript is too large (22 pages). The authors could switch less important images and materials/methods to suppl. Info.

Conclusion: The conclusion should be shortened (maximum 100 words)

Round 2

Reviewer 2 Report

Review to the revised manuscript entitled “The Discovery, Enzymatic Characterization and Functional Analysis of a newly isolated Chitinase from Marine-Derived Fungus Aspergillus fumigatus df347” by Ya-li Wu et al.

I sincerely thank the authors for their effort in careful revision of the manuscript. The manuscript content, readability and discussion of the results have been considerably improved during the revision. The issues and comments that were pointed out have been mostly addressed by the authors.

There are only some minor remarks and suggestions to consider before acceptance of publication.

The language and English grammar of the manuscript could still be improved. The use of professional editing service is encouraged.

The accordance of the formatting with the journal guidelines should be carefully checked.

The word “good” in phrases like “good performance” (line 18), “good stability (line 187), “good acid resistance” (line 190) is somewhat subjective and is not reflecting scientific use of language. For example, it is not clear what the authors mean by “good performance”. Is it used to reflect a high specific activity or high catalytic efficiency or even high productivity or stability of the enzyme? Instead of “showed good stability”, the authors could use for example “was stable”.   

Line 24. As the stimulatory effect of some metal ions and Tris was up to 1.7 fold, it is hardly “the greatest stimulatory effect” and the sentence could be rephrased.

Typographical error on line 88.

Line 94-95. The sentence is not clear. There seem to be some mistake in phrasing.

Lines 129, 162, 165, 168, 482. The “kilodalton” abbreviation should be kDa not KDa. The unit K means “Kelvin degrees” which is misleading.

Typographical error on line 235.

Lines 265-267. The affinities (Km values) cannot be compared with each other if one is presented in mM and other in mg/ml.

Lines 310-312. What was the enzyme content in the reaction mixture?

The quality/resolution of Figure 4 should be higher as currently the numbers on the peaks are almost not readable. The panels should be marked.

Chapter 2.11. For the amino acid abbreviations only one style (one or three letters) should be used throughout the text.

Chapter 3.5, 3.7., 3.9., 3.13., 3.14. Instead of rpm values the g force values should be presented.

Line 423, 426, 429. Should the unit be “µl” instead of “ul”? Line 559. Is “um” actually “µm”?

Supplementary material: The included additional file is very helpful to obtain the important information related to the study but the Table A2 should still be in the main text as the vital results are not provided in the main text otherwise.

Reviewer 3 Report

The paper has been improved and now can be accepted for publication.

Author Response

Dear Reviewer,

Thanks very much for taking your time to review this manuscript. I really appreciate all your comments and suggestions! 

Reviewer 4 Report

N/A

Author Response

(The authors gave the same response as above.)
